# Association of Blood Mercury Level with Liver Enzymes in Korean Adults: An Analysis of 2015–2017 Korean National Environmental Health Survey

**DOI:** 10.3390/ijerph20043290

**Published:** 2023-02-13

**Authors:** Jin-Wook Chung, Dilaram Acharya, Jitendra Kumar Singh, Joon Sakong

**Affiliations:** 1Division of Cardiology, Department of Internal Medicine, College of Medicine, Keimyung University Dongsan Medical Center, Daegu 41931, Republic of Korea; 2Department of Management, Evaluation and Health Policy, School of Public Health, Université de Montréal, Montréal, QC H3N 1X9, Canada; 3Medical Affairs and Innovation, Héma-Québec, Montréal, QC H4R 2W7, Canada; 4Department of Community Medicine, Janaki Medical College, Tribhuvan University, Janakpur 456000, Nepal; 5Department of Preventive Medicine and Public Health, College of Medicine, Yeungnam University, Daegu 42415, Republic of Korea

**Keywords:** alanine aminotransferase, aspartate aminotransferase blood mercury level, Korean adults

## Abstract

Increased liver enzymes as a result of exposure to mercury and their toxic effects are not well understood in Korea at the population level. The effect of blood mercury concentration on alanine aminotransferase (ALT) and aspartate aminotransferase (AST) was evaluated after adjusting for sex, age, obesity, alcohol consumption habit, smoking, and exercise parameters in 3712 adults. The risk of abnormal liver function was measured using a multiple logistic regression analysis. Blood mercury concentration was divided into quartiles, and liver enzyme levels were compared for each quartile. ALT and AST levels were 10–20% higher in the second, third, and fourth quartiles compared to the first quartile. The risk of liver dysfunction or elevated liver enzymes was significantly higher in the second, third, and fourth quartiles than in the first quartile. As blood mercury levels increased, liver enzymes and mercury-induced hepatotoxicity increased. The increase in liver enzymes caused by mercury was more pronounced in the low-mercury concentration range. To reduce the long-standing problem of abnormal liver enzymes and liver function in Korea and other similar settings, it is important to decrease exposure to mercury through effective implementation of specific health and environmental strategies.

## 1. Introduction

Mercury (Hg) exists in its elemental form as inorganic mercury, mercury compounds, and organic mercury compounds. The toxicity and health effects of these three compounds differ. The elemental and inorganic forms of mercury are mainly absorbed in the respiratory tract. However, organomercury is lipophilic, and is therefore mainly absorbed and bioaccumulated through the gastrointestinal tract [1,2]. Mercury is excreted from the body through the urine and feces. The half-life of the total mercury in the blood is about 57 days [3]. In Japan, the health effects of catastrophic mercury poisoning were widely known through Minamata disease. In Minamata disease, which is caused by the ingestion of mercury-contaminated seafood, neurological symptoms predominate [4]. Exposure to high concentrations of mercury affects the cardiovascular, endocrine, reproductive, and immune systems [5]. 

Many cases of mercury hepatotoxicity have been reported. There have been cases of acute poisoning in children exposed to mercury vapor of an unknown dose [6]. Hepatocellular effects include biochemical changes (e.g., serum alanine aminotransferase (ALT), ornithine carbamyl transferase, or an increase in serum bilirubin concentration) and decreased coagulation factor synthesis. Hepatic edema and central lobular vacuolation have been observed in men who died from acute exposure to high concentrations of metallic mercury [7,8]. The effects of oral exposure to inorganic mercury on the human hepatobiliary system are limited. A 35-year-old male who ingested a lethal dose of mercuric chloride developed jaundice and increased levels of aspartate aminotransferase (AST), alkaline phosphatase, lactate dehydrogenase (LDH), and bilirubin [9]. Hepatomegaly and softening were observed in his autopsy. Hepatomegaly was also observed in a 19-month-old boy who ingested an unknown dose of mercuric chloride powder [10]. Hepatobiliary effects from oral exposure to organic mercury are very limited. However, autopsies of four adults and four infants who died from methylmercury poisoning in Iraq in 1972 showed fatty liver changes [11]. Nevertheless, in an epidemiological study conducted in the Minamata area, the prevalence of liver disease in areas with methylmercury was not significantly higher than that in controls not exposed to methylmercury [12]. 

Mercury toxicity is caused by cell degeneration, oxidative stress, and alterations to energy metabolism [13,14]. Mercury increases reactive oxygen species (ROS) levels, leading to tissue damage due to oxidative stress, and induces peroxide production [15]. The increase in ROS production leads to a disruption of the antioxidant system defense and cell death due to lipid peroxidation [16]. Mercury induces the oxidation of DNA, proteins, and lipids in liver cells by increasing reactive oxygen levels [17]. It also destroys liver cell membranes, resulting in the cell contents leaking into the blood, and induces hepatocellular necrosis [18]. Mercury additionally denatures enzymes and inactivates them by binding to their sulfhydryl groups [13]. Mercury activates various metabolic pathways in hepatocytes, leading to the apoptosis of hepatocytes [19]. 

Biomonitoring of environmentally hazardous substances on a national scale is gradually expanding, including the United States’ National Health and Nutrition Survey (NHANES), the Canadian Health Measures Survey (CHMS), and the German Environmental Survey (GerES) [20,21,22]. In Korea, per Article 14 of the 2009 Environmental Health Act, the government is conducting a national environmental health survey to investigate the concentration and influential factors of harmful environmental substances. The Korean National Environmental Health Survey (KoNEHS) estimates the representative value of the Korean population’s exposure level to harmful environmental substances and compares this to other countries [23]. There are few studies on the hepatotoxicity of mercury in a population. A study in the United States using the National Health and Nutrition Examination Survey (2003–2008) reported that blood methylmercury levels were significantly associated with liver function [24]. 

In Korea, subclinical changes in liver function are associated with blood mercury levels [25]. In a study using an elderly cohort, liver enzyme levels, including aspartate aminotransferase (AST), alanine aminotransferase (ALT), and gamma-glutamyl transferase (GGT), were significantly associated with blood mercury levels [26]. The elevated liver enzymes group had higher mean blood mercury levels than the normal group, and elevated blood mercury was associated with a 35% higher risk of elevated liver enzymes [27]. Elevated blood mercury levels were strongly associated with nonalcoholic fatty liver disease in a non-obese population [28]. However, in a Korean study involving repeated measurements of blood mercury and liver enzymes, liver enzyme levels had no significant relationship with mercury, except in women who consumed alcohol [29]. Several studies have observed the relationship between body mercury levels and liver function within the population, with inconsistent results. This study is significant for understanding the existing facts in the current settings and could help to draw the attention of the government and other associated stakeholders to formulate countrywide preventive strategies to reduce the problem as well as reduce the gap in the literature. Thus, this study was conducted to evaluate the effect of blood mercury concentration on liver enzyme levels in Korean adults after adjusting for age, sex, body mass index (BMI), smoking, alcohol consumption habits, exercise, and fish consumption, using the KoNEHS (2015–2017).

## 2. Materials and Methods

### 2.1. Study Participants

In Korea, according to Article 14 of the Environmental Health Act, a KoNEHS has been conducted every three years since 2009. It measures the concentration of environmentally harmful substances in the human body and evaluates associated risk factors through a nationwide cross-sectional survey. The KoNEHS is designed to be representative of residence, gender, and age by recruiting approximately 2000 subjects annually through a stratified, multi-stage sampling unit. This study used data from the Korean National Environmental Health Survey (2015–2017) (KoNEHs Cycle 3). KoNEHS Cycle 3 is a monitoring program implemented by the National Institute of Environmental Research from 2015–2017 to measure the exposure level to environmental hazards, such as lead and mercury, in Korean people. There were 233 regions and 183 childcare or educational institutions across the country eligible to participate. A total of 6167 people had their blood and urine collected. The survey data additionally provide information on lifestyle, including questions about age, sex, education level, and smoking and alcohol consumption habits. Of the 6167 people who participated in the survey, 3787 adults 19 years of age or older were analyzed. Among them, 33 people who had liver disease and 42 patients with no blood mercury data were excluded from the study. Finally, 3712 adults were included in the sample. This study was approved by the Research Ethics Committee of Yeungnam University (7002016-E-2022-001-01).

### 2.2. Blood Mercury Level Measurements

In the KoNEHS, the blood samples were collected using metal-free EDTA polyethylene tubes (BD, Franklin Lakes, NJ, USA) and the urine samples were collected using 15 mL polystyrene conical tubes (SARSTEDT AG&Co., Numbrecht, Germany) [30]. The collected samples were then stored frozen at −20°C until analysis. The blood mercury concentration was analyzed by flow injection cold-vapor atomic absorption spectrometry (AAS) (DMA-80; Milestone, Bergamo, Italy). Accuracy measurement results used QC samples (CllinChek level I) and the mean (SD) of 1.50 (0.05), 1.33 (0.06), and 1.30 (0.06) μg/dL for 2015, 2016, and 2017, respectively, and the mean accuracy (SD) of 100.5 (3.5), 95.8 (4.6), and 91.7 (5.6). 

### 2.3. Liver Enzyme Measurements

Blood liver enzyme levels were measured in the KoNEHS by analyzing the serum concentrations of ALT and AST at a wavelength of 340 nm using a colorimetry (Colorimetry, Modified IFCC UV method, ADVIA 1800, Siemens, Munich, Germany) [31]. Abnormal cut-off points were defined as 35.0 IU/L or more for ALT and 40.0 IU/L or more for AST. Elevated ALT and AST were defined as median values of crude 24.0/26.0 IU/L or higher, adjusted 19.1/28.1 IU/L or higher for men, and crude 18.0/23.0 IU/L and adjusted 17.3/27.6 IU/L or higher for women, respectively.

### 2.4. Social and Lifestyle Characteristics

In this study, age, sex, BMI, smoking history, alcohol consumption, exercise, and frequency of fish intake were investigated. Age was divided into six groups: 19–29 years old, 30–39 years old, 40–49 years old, 50–59 years old, 60–69 years old, and over 70 years old. Body mass index (BMI, kg/m^2^) was classified into three groups: “23 or less”, “23 or more and 25 or less”, and “25 or more”. Smoking status was classified into three groups: “never smoked”, “past smoker”, and “current smoker”. Alcohol consumption and exercise were classified into “yes” and “no” groups, and the frequency of fish intake was divided as: “rarely”, “1 to 3 times a month”, “once a week”, and “more than twice a week”. 

### 2.5. Statistical Analysis

The multistage stratified cluster sampling method was applied for sampling [32]. The primary sampling unit is the survey district, and the secondary sampling units are the household and individual. In the sampling process, weights were calculated to reflect the difference in sampling probabilities by household that resulted from applying the multistage stratified cluster sampling method. Blood mercury (BHg), serum AST and ALT levels were compared across the groups using the t-test or analysis of variance depending on data types. The statistical significance of the difference in liver enzyme levels according to mercury concentration in the body was tested using a general linear model. Age, sex, smoking, alcohol consumption, obesity, and exercise variables were adjusted. A sample of 3712 adults was divided into quartile groups according to blood mercury concentrations, and we statistically analyzed differences in liver function test results (ALT, AST). The ALT and AST cut-off values for obtaining the odds ratio (OR) values were 35 IU/L and 40 IU/L, respectively, and the OR for abnormal ALT, AST, elevated ALT, and AST was measured for each quartile. To visually evaluate the changes in ALT and AST according to blood mercury concentration, LOESS (local regression) smoothing, which draws the most appropriate prediction line for each data section, and a regression line were expressed in a scatterplot, respectively. The results of adjusting for age, sex, smoking, alcohol consumption, obesity, and exercise and crude data are presented together. The statistical significance of the results was set at the 5% level. All statistics in this study are based on SPSS ver. 25.0 (SPSS Inc., Chicago, IL, USA). 

## 3. Results

Of 3712 patients, 1617 (43.6%) were male and 2095 (56.4%) were female. The majority of subjects were in the 60–69 age range, and a few subjects were in the 19–29 age range. In the cohort, 45.9% of men and 37.6% of women were overweight, 32.8% of men were current smokers, and 91.3% of men were drinkers, while 40.0% of men and 34.7% of women exercised. Nearly 30% of both men and women consumed fish twice or more per week, and the frequency of fish intake was very similar for men and women (Table 1). The blood mercury concentration was 3.28 µg/L for men which was statistically significantly higher than the 2.30 µg/L for women. AST and ALT were also statistically significantly higher in males than in females at 26.46 IU/L and 26. 27 IU/L, respectively (*p* < 0.001). The blood mercury concentration increased with age and started to decrease in those in their 60s. ALT increased with age until the 50–59 age range and decreased at age 60; AST increased until the 60–69 age range.

As BMI increased, blood mercury concentration, ALT, and AST tended to increase (*p* trend < 0.001). Serum mercury and liver enzyme levels were higher in current and past smokers than in the non-smoking group (*p* trend < 0.001). Blood mercury concentration was significantly higher in the exercise group (*p* < 0.001). As the frequency of fish intake increased, the concentration of mercury increased significantly (*p* trend < 0.001), whereas ALT and AST levels were not affected by the frequency of fish intake (Table 2).

After dividing the blood mercury concentrations into quartiles, we compared the ALT and AST concentrations of each group. Before and after adjusting for covariates (gender, age, smoking, alcohol consumption, obesity, and exercise), ALT concentrations in the second, third, and fourth quartiles were higher than those in the first quartile. However, AST was not affected by mercury levels. ALT before covariate adjustment was increased by 10–20% to 27.01, 25.29, and 27.56 IU/L in the second, third, and fourth quartile groups, respectively, compared to 22.42 IU/L in the first quartile group. Even after covariate adjustment, ALT increased from 19.59 IU/L to 22.85, 21.07, and 22.38 IU/L (Table 3) for the second, third, and fourth quartile groups.

An abnormal liver enzyme value was defined as 35 and 40 IU/L or more for ALT and AST, respectively. Elevated liver enzyme levels were defined when the median value was exceeded for men and women. In multiple logistic regression analysis, when the first quartile of blood mercury concentration was used as the reference group, the odds ratio of abnormal ALT and AST levels showed a tendency to increase significantly in the second, third, and fourth quartiles. The OR of elevated ALT and AST levels also increased significantly in the second, third, and fourth quartiles compared to the first quartile of blood mercury concentration. However, the odds ratio did not increase significantly within the second, third, and fourth quartiles of blood mercury concentrations. This phenomenon was similar in ALT and AST (Table 4).

The relationship between blood mercury concentration and liver enzymes was analyzed using scatter plots and LOESS lines. In the case of ALT, a blood mercury concentration of less than 4 µg/L showed an increase before and after adjustment, but there was no significant change thereafter. In the case of AST, the tendency to increase was not greater than that of ALT (Figure 1).

## 4. Discussion

Although the causes of liver disease are complex and there are a variety of diagnostic methods, AST and ALT are important enzymes for diagnosing liver disease. Recent studies report that these enzymes are associated with overall mortality or death from specific diseases [33,34]. Liver function tests help prevent disease progression through early detection of liver disease. Prevention of liver disease should occur through individual risk factor management rather than treatment. For the effective prevention and management of various liver diseases found during an individual’s health examination, it is necessary to identify the disease incidence rate and its risk factors and establish appropriate management measures according to the individual diagnosis. 

However, there have been many Korean studies on liver disease and lifestyle factors, but few epidemiological studies on heavy metals, including mercury, and liver disease. We found that blood mercury concentration has a significant effect on increasing liver enzymes, suggesting hepatotoxicity. Before and after adjusting for covariates such as gender, age, smoking, alcohol consumption, obesity, and exercise, ALT was higher in the second, third, and fourth quartiles than the first quartile of blood mercury concentration. Both crude ALT and adjusted ALT were 10–20% higher in the highest quartile group than in the lowest quartile of blood mercury concentration. In multiple logistic regression, when the first quartile of blood mercury concentration was used as the reference group, the odds ratio of abnormal liver enzymes significantly increased in ALT and AST in the second, third, and fourth quartile groups, except for adjusted AST. These results were similar to those of Lee et al. [25], which showed a significant relationship between blood mercury levels and liver enzymes after adjusting for potential confounders and that the odds ratios of having abnormal ALT levels were statistically significant in the group with the highest blood mercury concentrations. Our results were consistent with those of Yang et al. [28], who found that blood mercury levels had a positive correlation with nonalcoholic fatty liver disease. Our findings were similar to Lee et al. [25], who found that subclinical liver function changes were correlated with blood mercury levels. Lee et al. [27] reported that mercury exposure was associated with elevated liver enzymes in a study using the second cycle of the KoNEHS. 

ALT and AST increased by 10–20% as the blood mercury concentration gradually increased, but the increase slowed down at a certain blood mercury concentration. There may be levels of mercury in the blood that no longer cause severe hepatotoxicity. Compared to AST, ALT was significantly affected by blood mercury concentration. Subjects were divided into an abnormal liver function group and an elevated liver enzyme group, and we performed a multiple logistic regression. In both, the abnormal group and the elevated group, the OR of the second, third, and fourth quartiles significantly increased compared to the reference group. However, the OR also no longer increased significantly in the second, third, and fourth quartiles of blood mercury concentration. This phenomenon was similarly observed in ALT and AST. It is presumed that the increase in liver enzymes accompanying the increase in blood mercury concentration has a supralinear relationship, not a linear one. This phenomenon means that hepatocytes react sensitively to mercury at a low concentration rather than at a high concentration. This finding was revealed through this study. Koreans tend to have higher levels of mercury in their blood than people from other countries. 

The blood mercury concentrations of Korean adults were reported as 3.12 µg/L in cycle I, 3.17 µg/L in cycle II, and 2.75 µg/L in cycle III. The blood mercury concentration was slightly lower than 4.30 µg/L in the Korean National Health and Nutrition Examination Survey (KNHANES) [35], but higher than the 0.81 µg/L in the United States and 0.79 µg/L in Canada [20,21]. Compared with Germany’s HBM I blood mercury concentration of 5 µg/L [36], the 95th percentile of blood mercury in Korean adults was 9.00 µg/L in the KoNEHS Cycle 3, which is about twice that of HBM I. In the KNHANES, blood mercury levels in 2005, 2008, 2009, 2010, and 2011 declined to 4.19 μg/L (3.99–4.39), 4.73 μg/L (4.57–4.89), 4.25 μg/L (4.09–4.41), 3.64 μg/L (3.49–3.80), and 3.08 μg/L (2.95–3.22), respectively. The blood mercury concentration of Koreans is still higher than that of the Asian populations analyzed by the NHANES [37]. 

Korea is surrounded by sea on three sides, and seafood is therefore abundant in Korean diets. In some areas, residents eat large predatory fish, such as sharks, which leads to higher mercury concentrations [38,39]. We found that the group who ate fish more than twice a week had higher mercury concentrations than the group who ate less fish. These results are consistent with the National Health and Nutrition Examination Survey, which found that the blood mercury concentration of Korean adults is related to the frequency of fish and shellfish intake [40]. In a national biomonitoring survey conducted in the United States and Canada, fish consumption was also significantly related to an increase in blood mercury levels. Chronic heavy metal exposure causes serious health impairment in humans over time. These disorders interfere with the normal biological mechanisms of major organs such as the liver, heart, brain, and kidneys. Because these metals do not degrade naturally within ecosystems, a longer exposure poses long-term health risks [41]. The toxic effects caused by heavy metals are mainly caused by oxidative stress in cells due to the generation of reactive oxygen species (ROS) [42].

These ROS cause various diseases, such as inflammation-related damage to the liver, kidneys, and cardiovascular system. The harmful effects of ROS are often neutralized or inhibited by antioxidants, but heavy metals interfere with this mechanism, leading to toxicity [41]. Mercury increases peroxides, with exacerbating effects on the antioxidant system and lipid peroxidation. A change in the signaling of the rf2-Keap1 molecule has been noted. Genomic changes include DNA damage, while microstructural changes include an increase in mitochondrial volume, excessive hydrogen peroxide formation, RER expansion, and phosphorylation of JNK GRP78 protein activators [43]. 

Mercury also induces pyknotic, nuclear, and mitotic disorders in cells. Excessive mercury exposure induces inflammation by overexpression of NF-kB and iNO and increases cytokine expression. Activation of Kupffer cells leads to a proliferative state. Mercury toxicity disrupts cholesterol marker levels and downregulates the expression of Bcl-2. It also increases activation of certain caspases, upregulating p53 expression. Eventually, mercury causes apoptosis and cell necrosis [43]. In this study, the concentration of mercury in the blood was higher in males than in females and increased with age. Mercury concentration increased with increasing age between the 20s and 50s, but decreased after age 60 [44,45]. This is presumed to be because males and young adults consume more seafood than the elderly. Because this is a cross-sectional study, it was difficult to ascertain a clear causal relationship, as only the correlation of variables related to hazardous substances was confirmed. In addition, the investigation of various exposure factors to mercury was limited. In particular, because only total mercury in the blood was measured, organic and inorganic mercury exposure through consumption of seafood or amalgam could not be investigated. 

This study is valuable in terms of the identification of the level of mercury concentration leading to an increase in liver enzymes from a recent large dataset of the Korean National Environmental Health Survey. However, there are certain specific limitations of this study to be considered in the interpretation of the results. Liver enzymes are affected by a wide variety of factors besides mercury, including alcohol consumption habits. These habits significantly affected ALT in men. Additionally, it was found that exercise affected AST in women. Nevertheless, quantitative analysis was not sufficiently performed by classifying alcohol consumption habits and exercise variables only with “yes” and “no” categories. Another limitation is the fact that other factors that may impact liver function, such as alcohol consumption the day before the test, medications, test errors, liver dystonia, and liver and biliary system diseases, were not considered in this study.

## 5. Conclusions

This study demonstrated that obesity, smoking and alcohol consumption, as well as blood mercury levels, impact liver enzymes. After adjusting for age, BMI, alcohol consumption, smoking, and exercise, which were significant in the univariate analysis, the increase in liver enzymes and the risk of abnormal levels were higher in people with high mercury levels than in those with low mercury levels. Therefore, to reduce the incidence of abnormal liver enzymes and liver function which has been a long-standing problem in Korea and other similar settings, it is important to reduce exposure to mercury though effective implementation of specific health and environmental strategies. In the future, long-term follow-up studies should be conducted to clarify the relationship between blood mercury levels and abnormal liver enzymes. 

## Figures and Tables

**Figure 1 ijerph-20-03290-f001:**
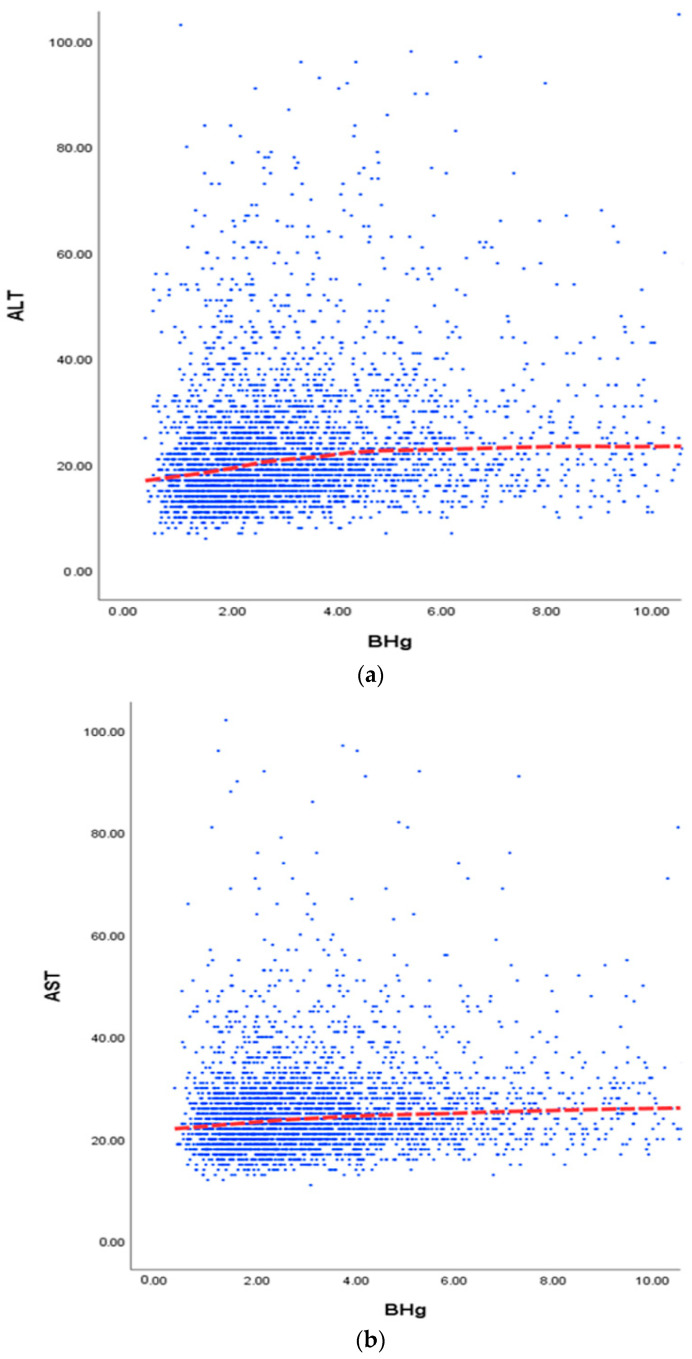
(**a**)**.** Relationship between blood levels of Hg (mercury). (**b**). Relationship between blood levels of Hg and liver enzyme (ALT) in adults (before adjustment). And liver enzyme (AST) in adults (before adjustment). (**c**). Relationship between blood levels of Hg and liver enzymes in adults (after adjustment). The smoothing line is fitted by using the LOESS regression model, adjusted for gender (in all), age, and smoking, alcohol consumption, obesity and exercise. BHg, Blood mercury; AST, aspartate aminotransferases; ALT, alanine transaminase.

**Table 1 ijerph-20-03290-t001:** General characteristics of study population by classification variables, KoNEHs 2015–2017.

Variables Classification	Male	Female	Total
Age (in years)			
19–29	135 (8.3)	142 (6.8)	277 (7.5)
30–39	220 (13.6)	314 (15.0)	534 (14.4)
40–49	271 (16.8)	339 (16.2)	610 (16.4)
50–59	271 (16.8)	339 (16.2)	610 (16.4)
60–69	406 (25.1)	509 (24.3)	915 (24.6)
70-	239 (14.8)	275 (13.1)	514 (13.8)
BMI (Kg/m^2^)			
<23	442 (27.3)	794 (37.9)	1236 (33.3)
≥23 and <25	433 (26.8)	514 (24.5)	947 (25.5)
≥25	742 (45.9)	787 (37.6)	1529 (41.2)
Smoking			
Never smoked	386 (23.9)	1989 (94.9)	2375 (64.0)
Past smokers	700 (43.3)	47 (2.2)	747 (20.1)
Current smokers	531 (32.8)	59 (2.8)	590 (15.9)
Alcohol consumption			
No	140 (8.7)	603 (28.8)	743 (20.0)
Yes	1477 (91.3)	1492 (71.2)	2969 (80.0)
Exercise			
No	970 (60.0)	1367 (65.3)	2337 (63.0)
Yes	647 (40.0)	728 (34.7)	1375 (37.0)
Fish intake			
Rarely	139 (8.6)	228 (10.9)	367 (9.9)
1–3/month	564 (34.9)	564 (34.9)	1316 (35.5)
1/week	424 (26.2)	424 (26.2)	938 (25.3)
≥2/week	490 (30.3)	490 (30.3)	1091 (29.4)
Total	1617 (43.6)	2095 (56.4)	3712 (100.0)

Data were expressed as number and frequency (categorical); KoNEHs, Korean National Environmental Health Survey; BMI, body mass index.

**Table 2 ijerph-20-03290-t002:** Mean and 95% confidence interval of blood mercury (µg/L), serum AST (IU/L) and ALT (IU/L) by characteristics of study population, KoNEHs 2015–2017.

Variables Classification	BHg	ALT	AST
Gender			
Male	3.28 (3.08, 3.50)	26.46 (25.28, 27.37)	26.27 (25.69, 26.87)
Female	2.30 (2.20, 2.41)	18.08 (17.61,18.56)	22.90 (22.46, 23.47)
*p*-value	<0.001	<0.001	<0.001
Age (in years)			
19–29	1.92 (1.72, 2.13) ^a^	19.25 (17.79, 20.83) ^a^	22.34 (21.01, 23.78) ^a^
30–39	2.88 (2.68, 3.10) ^c^	21.89 (20.75, 23.10) ^b,c^	23.27 (22.51, 24.06) ^a,b^
40–49	3.05 (2.87, 3.24) ^c,d^	22.45 (21.36, 23.59) ^c^	24.18 (23.47, 24.92) ^b^
50–59	3.24 (3.04, 3.45) ^d^	23.93 (23.00, 24.90) ^d^	26.16 (25.38, 26.96) ^c^
60–69	3.12 (2.90, 3.35) ^c,d^	22.75 (22.08, 23.45) ^c,d^	26.58 (25.90, 27.28) ^c^
70-	2.41 (2.16, 2.68) ^b^	20.55 (19.72, 21.41) ^a,b^	26.13 (25.41, 26.86) ^c^
*p*-value	<0.001	<0.001	<0.001
*p*-trend	<0.001	0.016	<0.001
BMI (Kg/m^2^)			
<23	2.31 (2.17, 2.45) ^a^	17.70 (17.10, 18.32) ^a^	22.85 (22.16, 23.55) ^a^
≥23 and <25	2.94 (2.76, 3.13) ^b^	21.65 (20.79, 22.54) ^b^	24.47 (23.82, 25.13) ^b^
≥25	3.11 (2.95, 3.28) ^b^	26.97 (25.97, 28.01) ^c^	26.39 (25.82, 26.97) ^c^
*p*-value	<0.001	<0.001	<0.001
*p*-trend	<0.001	<0.001	<0.001
Smoking			
Never smoked	2.39 (2.28, 2.51) ^a^	19.72 (19.16, 20.29) ^a^	23.48 (23.00, 23.98) ^a^
Past smokers	3.55 (3.33, 3.79) ^b^	25.67 (24.58, 18.32) ^b^	26.95 (26.01, 27.91) ^b^
Current smokers	3.37 (3.10, 3.65) ^b^	26.00 (24.49, 27.59) ^b^	25.90 (24.89, 26.95) ^b^
*p*-value	<0.001	<0.001	<0.001
*p*-trend	<0.001	<0.001	<0.001
Alcohol consumption			
No	2.43 (2.27, 2.60)	20.29 (19.13, 21.51)	24.01 (23.31, 24.73)
Yes	2.81 (2.68, 2.95)	22.14 (21.51, 22.78)	24.65 (24.15, 25.16)
*p*-value	<0.001	0.012	0.123
Exercise			
No	2.63 (2.50, 2.78)	21.76 (21.08, 22.46)	24.20 (23.67, 24.75)
Yes	2.95 (2.78, 3.14)	21.99 (21.21, 22.79)	25.16 (24.44, 25.89)
*p*-value	0.001	0.647	0.025
Fish intake			
Rarely	1.78 (1.54, 2.06) ^a^	20.59 (18.38, 23.07)	26.66 (22.73, 26.76) ^a,b^
1–3/month	2.58 (2.47, 2.71) ^b^	21.58 (20.84, 22.35)	23.89 (23.34, 24.45) ^a^
1/week	2.94 (2.75, 3.13) ^c^	22.15 (21.24, 23.01)	24.65 (23.92, 25.40) ^a,b^
≥2/week	3.39 (3.17, 3.62) ^d^	22.50 (21.71, 23.32)	25.3 6(24.75, 26.00) ^b^
*p*-value	<0.001	0.211	0.005
*p*-trend	<0.001	0.051	0.073
Total	2.75 (2.62, 2.88)	24.55 (24.1, 25.01)	21.84 (21.3, 22.4)

Values are presented as geometric mean (confidence interval). BHg, blood mercury; AST, aspartate aminotransferase; ALT, alanine transaminase. *p*-values and *p*-trend were calculated using the *t*-test or analysis of variance with values of BHg, ALT and AST. ^a,b,c,d^ No statistical difference between groups marked with the same letter in post hoc test.

**Table 3 ijerph-20-03290-t003:** Serum AST (IU/L) and ALT (IU/L) by quartile of blood mercury (µg/L) before and after adjustment of covariates #, KoNEHs 2015–2017.

Quartile	Crude GM (95% CI)		Adjusted GM (95% CI)	
BHg	ALT	AST	ALT	AST
Q1	22.42 (21.02, 23.81) ^a^	25.46 (22.83, 28.10)	19.59 (18.15, 21.02) ^a^	30.18 (27.53, 32.83)
Q2	27.01 (24.41, 29.62) ^b,c^	27.06 (25.32, 28.79)	22.85 (20.53, 25.16) ^b^	31.07 (29.33, 32.81)
Q3	25.29 (24.02, 26.55) ^b^	25.73 (24.81, 26.66)	21.07 (19.85, 22.30) ^a,b^	29.64 (28.77, 30.50)
Q4	27.56 (25.74, 29.37) ^c^	27.22 (26.29, 28.15)	22.38 (20.80, 23.95) ^b^	30.36 (29.48, 31.23)
*p*-trend	<0.001	0.370	0.020	0.478

# Covariates: sex, age, smoking, alcohol consumption, obesity, and exercise. *p*-trend were calculated using analysis of variance with value of BHg, ALT and AST. ^a,b,c^ No statistical difference between groups marked with the same letter in post hoc test. Quartile: first Q (≤2.240 for men, ≤1.689 for women), second Q (>2.241–3.446 for men, >1.690–2.420 for women), third Q (>3.446–5.440 for men, >2.421–3.650 for women), fourth Q (>5.441 for men, >3.651 for women). BHg, blood mercury; AST, aspartate aminotransferase; ALT, alanine transaminase; GM, geometric mean; CI, confidence interval.

**Table 4 ijerph-20-03290-t004:** Odds ratio and 95% CI of blood mercury (µg/L) for abnormal and elevated AST (IU/L) and ALT (IU/L) in logistic linear regression analysis before and after adjustment of covariates #.

Quartile BHg	Crude				Adjusted			
	Abnormal		Elevated		Abnormal		Elevated	
	ALT	AST	ALT	AST	ALT	AST	ALT	AST
Q1	Ref	Ref	Ref	Ref	Ref	Ref	Ref	Ref
Q2	1.60 (0.10, 2.56)	1.31 (0.89, 1.93)	1.64 (1.29, 2.09)	1.61 (1.25, 2.07)	1.54 (0.91, 2.63)	1.42 (0.99, 2.04)	1.40 (1.06, 1.84)	1.29 (0.98, 1.69)
Q3	1.32 (0.85, 2.04)	1.13 (0.72, 1.77)	1.71 (1.38, 2.11)	1.36 (1.04, 1.79)	1.13 (0.70, 1.83)	1.08 (0.77, 1.51)	1.16 (0.91, 1.49)	1.33 (1.05, 1.68)
Q4	1.80 (1.13, 2.88)	1.70 (1.15, 2.51)	2.12 (1.65, 2.73)	1.74 (1.34, 2.26)	2.06 (1.20, 3.51)	1.46 (1.02, 2.08)	1.33 (1.01, 1.75)	1.38 (1.07, 1.79)
*p*-trend	0.046	0.037	<0.001	<0.001	0.032	0.130	0.109	0.010

# Covariates: sex, age, smoking, alcohol consumption, obesity, and exercise. Quartile: first Q (≤2.240 for men, ≤1.689 for women), second Q (>2.241–3.446r men, >1.690–2.420 for women), third Q (>3.446–5.440 for men, >2.421–3.650 for women), fourth Q (>5.441 for men, >3.651 for women). Abnormal odds ratios; crude >35.0 IU/L for ALT, and >40.0 for AST. Elevated ALT and AST; over the median values of crude 24.0/26.0 IU/L, adjusted 19.1/28.1 for men, and crude 18.0/23.0 IU/L, adjusted 17.3/27.6 for women, respectively. BHg, blood mercury; AST, aspartate aminotransferase; ALT, alanine transaminase; Ref, reference group. *p*-trend were analyzed using the test of trend of odds.

## Data Availability

Not applicable.

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
