# Peer review of "Association of Blood Mercury Level with Liver Enzymes in Korean Adults: An Analysis of 2015–2017 Korean National Environmental Health Survey"

_ijerph, 2023, doi:10.3390/ijerph20043290_

Round 1
Reviewer 1 Report
Jin-Wook Chung and collaborators have carried out a clinical study in which they have verified that blood mercury levels in the Korean population are not only higher than those of other countries in the world, but also that these high levels could be responsible for the development of liver disorders.
From my point of view, it is a well-developed study and well described in the manuscript. The introduction includes everything necessary to centralize the topic, the methodology is correct, the results and the discussion are coherent and well presented; and the conclusions are appropriate to the findings obtained.
I just have a few minor suggestions that I think should be considered before the final publication of the article:
- In the abstract, some introductory lines should be included.
- Some abbreviations need to be defined in the text. For example:
* Line 124: QC
* Line 97: BMI
* All abbreviations for tables and figures should be defined in the corresponding figure/table captions.
- The paragraph on lines 124-127 should be rewritten so that it can be understood better (I think there is a wording problem).
- The paragraph on lines 131-135 should be rewritten so that it can be better understood (I think you mean that "the selected cut-off points from which the odds ratios were calculated were...", is that correct?
- In the statistical analysis section, the tests used to make the comparisons between the groups are not specified.
- Line 165, the comma must be added to the number 1617.
Author Response
Thank you very much for your comments and suggestions as a result of which we got a chance to improve our manuscript. In this revised version of the manuscript, we have marked with blue coloured writing for all changes made according to reviewers’ comments and suggestions to allow their verification. In addition, we have made point-to-point clarification of the reviewers’ comments and suggestions as given below:
- In the abstract, some introductory lines should be included.
Response: Agree. Corrected.
- Some abbreviations need to be defined in the text. For example:
* Line 124: QC
* Line 97: BMI
* All abbreviations for tables and figures should be defined in the corresponding figure/table captions.
Response: Agree. Corrected.
- The paragraph on lines 124-127 should be rewritten so that it can be understood better (I think there is a wording problem).
Response: Agree. Corrected.
- The paragraph on lines 131-135 should be rewritten so that it can be better understood (I think you mean that "the selected cut-off points from which the odds ratios were calculated were...", is that correct?
Response: Agree. Corrected.
- In the statistical analysis section, the tests used to make the comparisons between the groups are not specified.
Response: Agree. Corrected.
- In line 165, the comma must be added to the number 1617.
Response: Agree. Corrected.
Reviewer 2 Report
The summary should contain some conclusion of the research conducted.
Material&Methods: authors should indicate the reagents used with the brand and country of origin.
Conclusions are very brief, explain a little more based on the objectives of the study.
Author Response
Thank you very much for your important comments and suggestions. In this revised version of the manuscript, we have marked with blue-coloured writing for all changes made according to reviewers’ comments and suggestions to allow their verification. In addition, we have made point-to-point clarification of the reviewers’ comments and suggestions as given below:
Response: Agree. Corrected brand name and country of origin for suggested reagents. We also have revised the conclusion with more recommendations based on our findings.
Reviewer 3 Report
This study evaluated the effect of blood mercury concentration on alanine aminotransferase and aspartate aminotransferase after adjustment for sex, age, obesity, alcohol consumption, smoking, and exercise in 3,712 adults. The risk of abnormal liver function was measured by multiple logistic regression analysis. The Authors state that as blood mercury levels increased, liver enzymes and mercury-induced hepatotoxicity increased. In addition, mercury-induced liver enzyme elevations were more pronounced in the low mercury range.
There are certain specific limitations of this study to be considered in the interpretation of the results. Liver enzymes are affected by a wide variety of factors besides mercury, including alcohol consumption habits. These habits significantly affected ALT in men. Also, it was found that exercise affected AST in women. Quantitative analysis was not sufficiently performed by classifying drinking habits and exercise variables only with 'yes' and 'no' categories. Another limitation is other factors that may impact liver function, such as alcohol consumption the day before the test, medications, test errors, liver dystonia, and liver and biliary system diseases were not considered in this study. The Authors have correctly inserted these limitations in the manuscript.
This study shows interesting data and is well structured. The introduction provide sufficient background. The methods are clear and the statistical analysis was also well detailed. The conclusions are adequately supported by the results and interpretation of the data. The 4 tables are schematic but clear. The references are appropriate and current.
A few minor issues should be addressed:
· The Authors should correct the legends of figures 1A and 1B, probably the phrases "and liver enzyme" in the following line by mistake.
· The three images in figure 1 are not very sharp, mainly in my opinion because they are very small.
· English language minor spell check required.
Author Response
Thank you very much for your comments and suggestions and appreciation of our work. In this revised version of the manuscript, we have marked with blue-coloured writing for all changes made according to reviewers’ comments and suggestions to allow their verification. In addition, we have made point-to-point clarification of the reviewers’ comments and suggestions as given below:
Response: Thank you very much for your comments and suggestions. We agree with you that Figure 1A seems less sharp. For your kind information, we have checked and corrected spelling, grammar and other errors throughout the manuscript as suggested.